Production of cytotoxic compounds in dedifferentiated cells of Jatropha curcas L. (Euphorbiaceae)

Ovando-Medina Isidro 1
Pérez-Díaz Leny P. 1 2
Ruiz-González Sonia 1
Salvador-Figueroa Miguel 1
Urbina-Reyes Marcos E. 1
Adriano-Anaya Lourdes maria.adriano@unach.mx 1
1 Instituto de Biociencias, Universidad Autónoma de Chiapas , Tapachula , Chiapas , Mexico
2 Current affiliation:  Instituto de Investigaciones Biomédicas, Unidad de Investigación Periférica, Laboratorio de Virus y Cáncer, Instituto Nacional de Cancerología , Mexico
Day David
Electronic publication date: 2016 Nov 1
Publication date: 2016
Volume: 4
Electronic Location ID: e2616
Received 2016 Apr 29; Accepted 2016 Sep 27
Copyright: ©2016 Ovando-Medina et al.
Copyright year: 2016
Copyright holder: Ovando-Medina et al.
License: This is an open access article distributed under the terms of the Creative Commons Attribution License, which permits unrestricted use, distribution, reproduction and adaptation in any medium and for any purpose provided that it is properly attributed. For attribution, the original author(s), title, publication source (PeerJ) and either DOI or URL of the article must be cited.
License URL: https://creativecommons.org/licenses/by/4.0/

Keywords: Callus, Chromatography, Jatrophone, Terpenes, Cancer

Funding: The authors received no funding for this work.

==============================
This study addresses the in vitro culture as an alternative to obtain compounds with cytotoxic activity from the medicinal plant Jatropha curcas (Euphorbiaceae). We determined the presence of cytotoxic compounds in both whole plants and dedifferentiated cells. We evaluated the effect of auxin, cytokinins and light on callus induction in cotyledon explants. We found that the most effective combination to induce callus was the auxin 2,4-D (5 mM) with the cytokinin 6-BAP (2.5 mM), on Murashige-Skoog medium in darkness. We compared the callogenic potential among accessions from different geographic origins, finding that ARR-251107-MFG7 is most prone to form callus. The roots of J. curcas grown in field produced a compound chromatographically similar to the cytotoxic diterpene jatrophone. The profile of compounds extracted from the dedifferentiated cells was similar to that of the whole plant, including a relatively abundant stilbene-like compound. This study contributes to the future establishment of protocols to produce anti-cancer compounds from J. curcas cultivated in vitro.

Introduction

The search for less toxic and more potent anti-carcinogenic drugs is based on the fact that current drugs are scarcely selective and highly toxic to normal cells (Mohan, Koushik & Fuertes, 2012). The most promising sources of such molecules are plants, especially those used for herbal medicine (Alonso-Castro et al., 2011). In this sense, Jatropha curcas, a plant native to Mesoamerica and recently rediscovered due to its potential as a source of raw material for biofuels (Salvador-Figueroa et al., 2015), it has been used in traditional medicine by various peoples of Asia, Africa and Latin America (Kumar & Sharma, 2008). A variety of compounds with different biological activities have been isolated from species belonging to the genus Jatropha (Devappa, Makkar & Becker, 2010). Among them are: jatrophol, a molecule with rodenticide activity (Jing et al., 2005), the curcusones a, b and c, and jatropholone A, with antineoplastic properties (Muangman, Thippornwong & Tohtong, 2005; Naengchomnong et al., 1986; Naengchomnong, Tarnchompoo & Thebtaranonth, 1994; Theoduloz et al., 2009), hydroxyl-jatrophone and other diterpenes with potential antimetastatic activity (Devappa, Makkar & Becker, 2011). There is evidence suggesting that methanol extracts of the leaves have anti-metastatic and anti-proliferative activity (Balaji et al., 2009). That compounds have been isolated from different plant parts in several species of Jatropha, although the concentration is usually low (Goulart et al., 1993). This situation is not unique to this genus, as in other plants bioactive molecules are also found in low concentrations.

Given the above, the in vitro culture of dedifferentiated plant cells is an alternative for increasing the concentration of the compounds of interest (Roberts, 2007). In this regard, Fett-Neto et al. (1994) obtained 100 times more taxoid in Taxus cuspidata callus than in the field plant. However, in vitro culture does not always improve the concentration of the metabolite of interest (Pletsch & Charlwood, 1997), given the difficulties to obtain friable callus, the genetic variations throughout the culture and the formation of cell aggregates (Chattopadhyay et al., 2002). Therefore, the objectives of this study were (a) to establish a procedure for obtaining friable and fast growing calluses, and (b) to evaluate the production of cytotoxic compounds in J. curcas dedifferentiated cells.

Materials and Methods

Biological materials

Five accessions of J. curcas (Table 1) representing the regions of Chiapas (Mexico) were used, from the Institute of Biosciences (IBC, initials in Spanish) Jatropha Germplasm Bank of the Autonomous University of Chiapas (Mexico) located in the municipality of Tapachula, Chiapas (14.4976N, 92.4774W, 58 m a.s.l., annual average temperature 30.7 °C annual average humidity 80%, average rainfall of 2632.9 mm and andosol-type soil.) For in vitro culture, 50 seeds of each of the accessions were collected. For the whole plant phytochemical analysis, samples of leaf, stem and root of the accession MAP-011107-G8 were used. In a parallel study (I Ovando-Medina, 2016, unpublished data), that accession was the most toxic among many Mesoamerican J. curcas accessions analyzed. Those samples were washed with tap water, dried at 60 °C for 48 h and ground to particle size of 500 µm.

Table 1 Biological material used in this study.

Jatropha curcas accessions representative of regions in Chiapas State (Mexico) used in this study and their callogenic ability when induced with 2,4-D (5 µM) and BAP (2.5 µM).

Accession*	Latitude	Longitude	Population	Callus dry weight (mg)**	
ARR-251107-MFG7	16°11.231′	93°54.516′	Isthmus	1014 ± 578a	
MAP-011107-G8	15°25.505′	92°53.554′	Soconusco	402 ± 23ab	
JIQ-090208-AG1	16°40.012′	93°39.242′	Center	175 ± 31b	
PUJ-030 508-S4	16°16.430′	92°17.550′	Frailesca	204 ± 19b	
CDCU-030208-F4	15°40.473′	92°00.129′	Border	207 ± 50b	
Notes.

* Populations located in Chiapas, Mexico. Source: Ovando-Medina et al. (2011).

** Coefficient of variation was 64.9%. The data were taken 30 days after culture. Each accession had three replicates and each repetition consisted of 20–30 cuttings of a cotyledon. Different letters denote statistical differences revealed by ANOVA and Tukey’s tests (p ≤ 0.05). Composition of medium: Medium MS (1962), 3% sucrose p/v and 500 mg ⋅ L of Polyvinylpyrrolidone + 2,4-D (5 µM) + BAP (2.5 µM).

Induction of dedifferentiated tissue

Cotyledons of different J. curcas accessions were used as explants for induction of dedifferentiated tissue. In the first phase, the seeds of the accession MAP-011107-G8 were sown on MS medium (Murashige & Skoog, 1962), after disinfection with sodium hypochlorite at 5%, following the procedure described by Soomro & Memon (2007), and kept in 2 d darkness and 2 d in light. After that period, the seeds were cut transversely, the embryo was removed and cotyledons were sown on a MS medium supplemented with different hormone combinations, and under different lighting conditions. For this phase, we used a full-randomized design with 32 treatments including a control without hormones, with three replications. Explants were maintained for 20 d, at the end of which the dry weight of callus generated was quantified. Based on the treatment that induced the highest amount of callus, the optimization process was conducted based on the concentration of hormones, using a 62 factorial design, where the factors were the hormones (2,4-D and BAP) at six levels each, with four replications. In these treatments the dry weight of callus was determined after 30 d of culture. Lastly, cotyledons of all accessions were placed under the best conditions to induce callus, comparing among accessions.

Determination of jatrophone content in field plants

Three grams (±0.1 g) of particles of different plant parts were subjected to extraction in triplicate using refluxing (60 °C, 20 cycles) with 80 mL hexane in Soxhlet equipment. The hexane was evaporated in a rotary evaporator to 50 °C and the yield (w/w) was calculated. The separation and identification of jatrophone was performed by thin layer chromatography using silica gel 60 plates of 5 × 20 cm (Sigma-Aldrich®, Fluka, Germany) washed with MeOH (purity 99.8%; Hycel, Guadalajara, Mexico) activated at 50 °C for 5 min. For this, the residue obtained as previously described was dissolved in hexane to achieve concentrations of 0.1 g/mL. An aliquot (15 µL) of each of the extracts and of a mixture of jatrophone (10 mM) with jatropholone a and b (4 mM based on jatropholone a dissolved in Hexane: Ethyl Acetate 7:3, kindly provided by Dr. G Schmeda-Hirschmann of the University of Talca, Chile), were placed individually on the chromoplate lanes. The chromatogram was developed at 28 °C as a mobile phase a mixture of Hexane: Ethyl Acetate 7:3. The compounds were revealed with sulphuric anisaldehyde using the procedure reported by Pertino et al. (2007). Under the above conditions, the Rf values for jatrophone and jatropholone a and b mixture were 0.772 and 0.817 (given that isomers are not separated with this eluent mixture), respectively.

The positive hexanic extracts for jatrophone were subjected to column chromatography packed with silica gel 60 (Merck, Mexico City, Mexico). The column preparation and elution was performed using the method proposed by Goulart et al. (1993) with flow of 0.6 mL/min. The eluate was received in 2 mL fractions and the presence of jatrophone was verified by thin layer chromatography. Fractions where the compound of interest was present were combined, then the solvent was evaporated in an oven at 50 °C and the residue was dissolved in 1 mL methanol (purity 99.9%; J.T. Baker®, Mexico City, Mexico).

The solids dissolved in MeOH from the previous phase were analyzed on a gas chromatograph (Focus-GC, Thermo Fisher, Milan, Italy), coupled to a mass spectrum (MS DSQ-II; Thermo Fisher, Milan, Italy). The chromatography was performed at an intermediate polarity column (5ms SQC, Thermo Fisher, Milan, Italy) of 30 m × 0.25 mm, D.I. 0.25 × 0.25 µm using helium as gas carrier. The temperatures in the column of the injector and the ionization chamber used were based on those reported by Wang et al. (2009). The standards were analyzed similarly, separately. The chromatographic and spectrometric data were processed by the Xcalibur data system (Version 2.0.7, 2007). The fragmentograms obtained from each of the compounds present in the extracts were compared with those stored in the NIST 02 (2005) database.

Determining metabolites in dedifferentiated cells

Of each of the accessions, 2 g of callus (dry weight) were taken and subjected to an extraction and semi-purification process, following the procedure described for evaluating jatrophone content in field plants. We performed two sequential extractions, the first one with hexane and then the residue with ethanol 96° GL (Goulart et al., 1993). Extracts were analyzed by GC-MS, as described previously.

Data analysis

The concentrations of jatrophone and jatropholone a and b were obtained based on the areas under the curve of the samples peaks in relation to those of the standards. The abundance of an abundant stilbene-like compound (1) in callus was estimated using the peak’s area of the chromatogram. Data from all assays were subjected to analysis of variance (ANOVA) and comparison of means (Tukey α ≤ 0.05).

Results

Effect of auxins and cytokinins on the induction of callus

All hormone treatments induced callus formation in cotyledons of J. curcas, accession MAP-011107-G8. In all cases the callogenesis started on the edge where the cutting was made and then covered the rest of the explant. Table 2 shows the callus induction by the different phytohormone treatments and lighting conditions. It was observed that exogenous phytohormones under any lighting condition did not influence the formation of callus, as they were statistically similar to the control (p ≤ 0.05). The auxin 2,4- dichlorophenoxyacetic acid (2,4-D) presented lower values of callus formation, but its callogenic effect was potentiated by combining it with the cytokinin 6-benzylaminopurine (BAP), either under light or darkness. Besides this, it was observed that calluses from treatments with light showed signs of differentiation, producing chlorophyll and compacting themselves, while those cultivated in darkness remained undifferentiated and were friable (Fig. 1).

Table 2 Effect of phytohormones and lighting conditions on callus dry weight (mg) obtained from cotyledon explants of Jatropha curcas.

Each treatment was repeated three times and each replicate consisted of 20–30 cuttings of a seed cotyledon from accession MAP-011107-G8. Means with different letters are significantly different (p ≤ 0.05). Formulation of Medium: Basal Medium MS + 3% sucrose (p/v) + 500 mg ⋅L−1 PVP. Auxins: 2,4-D (2,4-dichlorophenoxyacetic acid), NAA (naphthalene acetic acid), IAA (indole-3-acetic acid). Cytokinins: KIN (kinetin), BAP (6-benzylaminopurine), SAD (adenine sulfate)

Treatments	Lighting	
	Light	Darkness	
Control	189 ± 8abc	183 ± 37abc	
2,4-D	135 ± 36c	136 ± 14c	
NAA	168 ± 18abc	190 ± 42abc	
AIA	178 ± 12abc	201 ± 30abc	
KIN	166 ± 36abc	152 ± 40bc	
BAP	168 ± 12abc	207 ± 33abc	
SAD	154 ± 37bc	128 ± 8c	
2,4-D + KIN	199 ± 33abc	235 ± 52abc	
2,4-D + BAP	289 ± 107a	280 ± 13ab	
2,4-D + SAD	154 ± 30bc	234 ± 39abc	
NAA + KIN	163 ± 18abc	245 ± 34abc	
NAA + BAP	170 ± 42abc	239 ± 95abc	
NAA + SAD	183 ± 23abc	208 ± 41abc	
AIA + KIN	197 ± 46abc	203 ± 52abc	
AIA + BAP	156 ± 37bc	205 ± 34abc	
AIA + SAD	227 ± 18abc	201 ± 46abc	

Figure 1 Influence of illumination on the appearance of Jatropha curcas calli.

Callus induced in conditions of light and darkness from cotyledon explants of Jatropha curcas. (A) Developing photosynthetic callus. (B) Developing callus of “sugary” and friable appearance.

To obtain the appropriate callus induction, different concentrations of phytohormones 2,4-D and BAP were evaluated under darkness. In the overall analysis of the treatments, the combinations of 2,4-D and BAP at 7.5 + 2 µM and 5 + 2.5 µM produced the largest amount of callus, were statistically different from the other treatments, but equal among them (Table 3). Statistically significant differences were found among the amounts of cell callus formed in the five accessions evaluated (Table 1). The ARR-251107-MFG7 accession was superior in its capacity of callogenesis.

Table 3 Combined effect of the auxin 2,4-dichlorophenoxyacetic acid (2,4-D) and cytokinin 6-benzylaminopurine (BAP) on the formation of cell callus in cotyledons of Jatropha curcas accession MAP-011107-G8.

The data were taken 30 days after culture in darkness. Each treatment had three replicates and each repetition consisted of 20 to 30 cuttings in a cotyledon. Different letters denote statistical differences revealed by ANOVA and Tukey’s tests (p ≤ 0.05). Formulation of Medium: Basal Medium MS + 3% sucrose (p/v) + 500 mg ⋅L−1 PVP.

BAP (µM)	2,4 D (µM)	
	0	1	2.5	5	7.5	10	
0	196 ± 10efghij	138 ± 10j	145 ± 7ij	186 ± 20ghij	142 ± 10j	191 ± 20efghij	
0.5	194 ± 10efghij	271 ± 110bcdefgh	153 ± 21hij	156 ± 30hij	194 ± 20efghij	208 ± 40defghij	
1	152 ± 20hij	199 ± 20defghij	246 ± 31cdefghij	285 ± 30abcdefg	303 ± 10abcdefg	313 ± 20abcde	
1.5	140 ± 20j	0.190 ± 20fghij	376 ± 24ab	223 ± 20cdefghij	245 ± 30cdefghij	267 ± 30bcdefghi	
2	203 ± 60defghij	0.221 ± 40abc	342 ± 46cdefghij	209 ± 60defghij	405 ±  11a	319 ± 20abcd	
2.5	211 ± 10defgihj	0.219 ± 30defghij	375 ± 33ab	402 ±  20a	311 ± 10abcdef	307 ± 20abcdefg	

Cytotoxic compounds in mother plants

The yield of the hexane extracts depended on the type of tissue of the J. curcas studied. The highest extract concentration was found in the roots (50 ± 2.7 mg/g dry weight), which had 1.6 1.8 times more extract than in the leaves (30 ± 2.8 mg/g dry weight) and bark (27.1 ± 3.9 mg/g dry weight). Separation by thin layer chromatography of the components in all the hexane extracts showed a compound with Rf similar to jatropholone (0.814), while for jatrophone (Rf = 0.772) only one spot with similar Rf (0.768) was found in the root extract (Fig. 2). Separation and analysis by gas chromatography-mass spectrometry of jatrophone positive fractions revealed that the root hexane extract contains, in addition to other compounds, jatrophone and jatropholones a and b (Table 4). Fig. 3 shows the chromatogram and mass spectrum of the fraction in which jatrophone was identified.

Figure 2 Thin layer chromatogram of crude hexane extracts from different tissues of Jatropha curcas MAP- 011107-G8.

Extracts were diluted to 0.1 g/mL and revealed with sulphuric anisaldehyde (Pertino et al., 2007). Lanes: (1) mixture of standards Jatrophone 10 mM + Jatropholone at 4 mM + b; (2) leaf extract; (3) bark extract; (4) root extract. In lane 1 band “a” represents the jatropholones (Rf = 0.817); and band “b,” the jatrophone (Rf = 0.772).

Production of compounds in dedifferentiated cells

Although 19 compounds (alkanes, fatty acids, among others) were found under the assayed conditions of gas chromatography- mass spectrometry, no jatrophone or jatropholones were detected in dedifferentiated cells. A major stilbene-like compound (1) was found, which was present in all extracts with retention time of 27.21 min (Fig. S1) and whose fragmentogram showed the following ions and relative abundances: [280 (8), 279 (12), 273 (14), 167 (53), 166 (14), 165 (19), 163 (62), 150 (52), 149 (32), 148 (40), 147 (32), 145 (100), 112 (13), 110 (13), 101 (7), 83(11), 70 (40), 68 (24), 57 (19), 53 (19)].

There were no differences in the amount of (1) produced by calluses cultivated in the light or darkness. It was found that this metabolite is dependent on genotype as the accession ARR-1251107-MFG7 had the greatest peak’s area of this substance in comparison with the remainder. Compared to the roots of field plants, calluses had 26 times more (1).

Table 4 Compounds identified in the root hexane extract of Jatropha curcas accession MAP-011107-G8, by gas chromatography-mass spectrometry.

Yields were calculated based on the areas under the peak curve of purified standards.

Structure	Molecular Weight (g gmol−1)	Retention time (min)	Yield (mg compound g sample−1)	
Jatrophone	312	20.57	2.038	
Jatropholone a	295	22.1	6.331	
Jatrotropholone b	296	22.3	1.668	

Figure 3 Analysis of hexane extracts of Jatropha curcas by GC-MS.

(A) Chromatogram of a fraction of the root hexane extract of Jatropha curcas MAP-011107-G8, where the peak corresponding to jatrophone is circled. (B) Fragmentogram of jatrophone–retention time 20.57 min, molecular weight is circled 312g ⋅ gmol−1.

Discussion

Although a variety of compounds with cytotoxic activity isolated from J. curcas has been reported (Misra & Misra, 2010; Ravindranath et al., 2004; Van den Berg et al., 1995), our study focused on three compounds: jatrophone, jatropholones a and b which were present, especially in the roots. In particular, jatrophone is a diterpene present in the genus Jatropha, which had not been previously reported in J. curcas. Unlike the findings by Goulart et al. (1993), who detected jatrophone in J. elliptica using hexane as eluent in column chromatography, in this study it was detected in fractions where the elution solvent polarity was increased (Hexane: Ethyl Acetate 6:1).

Further studies of the extracts are required to corroborate the presence of jatrophone and discard the possibility that we are dealing with other terpenoid compounds with similar chemical structure. However, comparison of the mass spectrum of the compound detected [312 (14), 297 (4), 284 (29), 216 (36), 202 (33), 187 (34), 175 (45), 173 (83), 159 (100), 147 (30), 145 (70), 133 (30), 130 (45), 119 (30), 107 (35), 105 (56), 91 (32), 79 (40), 77 (50), 69 (35)] with that of the jatrophol, a terpenoid with the same molecular weight (Naengchomnong, Tarnchompoo & Thebtaranonth, 1994), showed that they have different ionization patterns [312 (100), 297 (11), 281 (43), 253 (34), 240 (37), 225 (35)]. Conversely, we emphasize that the compound identified as jatrophone has a ionization pattern very similar to that of the standard [312 (15), 297 (6), 241 (67), 189 (66), 187 (34), 175 (31), 173 (43), 160 (75), 159 (100), 147 (41), 145 (38), 115 (39), 91 (43), 81 (51), 79 (43), 53 (67)] and to that reported by Pertino et al. (2007) for jatrophone.

In this work a maximum of 2,038 µg/g of jatrophone was found, which represents 22.6 times the maximum reported for J. gossypiifolia (90 µg/g, Kupchan et al., 1976) and 1.4 times than in J. isabelli (1,450 µg/g, Dos Santos & Goulart Sant’ana, 1999), but 4.9 times less than in J. elliptica (10,000 µg/g, Goulart et al., 1993). The J. curcas accession MAP-011107-G8 has a high amount of jatrophone, although it is possible that the yields are affected by the extraction method. Although these results are promising with respect to other species, it must be remembered that in all cases the extraction is performed at the root, which involves waiting for the development of the latter and the complete sacrifice of the plant.

In regard to the study of induction for dedifferentiation, it was found that when the exogenous phytohormones auxin and cytokinin were at low concentrations did not influence the formation of cell callus. The formation of callus in the control treatment was remarkable (Table 2). The results of this study show that explants of J. curcas tend spontaneously to in vitro callogenesis, denoting a high endogenous hormone load or high responsiveness of the plant tissues to healing, since callogenesis invariably started at the cutting sites of the explant. In this sense, Sujatha, Makkar & Becker (2005) mention that J. curcas leaf explants develop callus from the cutting margins. In other species, in vitro spontaneous callogenesis has been observed, which has been linked either to high hormone concentrations or to its de novo synthesis. In order to determine the cause of spontaneous callogenesis in J. curcas, studies are needed regarding the dynamics of hormone concentration in intact tissues and in vitro cultured.

Although auxin 2,4-D presented the lowest values of callus formation, its effect was potentiated when combined with cytokinin BAP. For this reason wider ranges of concentration of both hormones were explored and found that at higher concentrations (greater than 2.5 µM in the case of auxins and above 2 µM in the case of cytokinins) there is an increased production of callus. Similar results were obtained by Kaewpoo & Te-Chato (2009) who evaluated different growth regulators in epicotyl and hypocotyl explants of J. curcas finding friable, soft and slightly yellow callus in all treatments with 2,4-D (6.78 µ M), while compact callus or shoots were obtained with the other auxins evaluated (IBA or NAA). In the present study friable callus induction was observed when using NAA although in lesser amounts, which is consistent with results obtained by Shrivastava & Banerjee (2008) who at assessing concentrations from 0.2 µM to 5.37 µM of NAA obtained the highest amount of callus at the highest concentration.

In the study for optimization of concentrations, it was observed that the largest amount of callus is formed at concentrations of 5 µM and 7.5 µM of 2,4-D, which are statistically equal (Table 3). However, we suggest the use of the lowest effective concentration, since high concentrations of auxin are almost always detrimental to the production of secondary metabolites (Kim et al., 2007).

In the case of the selection of cytokinins it is advisable to use BAP at 2.5 µM, since when combined with auxin callogenesis is potentiated, which did not occur with other cytokinins. This coincides with the study of Kalidass, Mohan & Daniel (2010). They studied the effect of BAP in the production of Catharanthus roseus callus and found that increasing the levels of BAP resulted in higher dry weight of callus. In other studies with explants of J. curcas effective concentrations of BAP for callogenesis were found, similar to the findings in the present study (Datta et al., 2007; Rajore & Batra, 2007; Wei et al., 2004).

In regard to lighting conditions, it does not influence the amount of callus formed, however, it was significant in the differentiation of callus tissue. Since the long-term objective is obtaining dedifferentiated cell cultures that can be maintained in bioreactors, the condition of darkness is preferable. Paz et al. (2006) mention something similar. They found that under lighting, friable calluses could be conditioned to a vitrification process, which they did not observe in the conditions of darkness. Also, Pletsch & Charlwood (1997) found that the production of jatrophone in cultures in vitro of J. elliptica decreases up to 15 times under continuous lighting. There are other studies which conclude that in the condition of darkness calluses can accumulate more secondary metabolites compared to those grown under light (Banthorpe et al., 1986; Mukundan & Hjortso, 1991; Yazaki et al., 2001).

It is worth noting that jatrophone was found in the mother plant but not in the calluses, probably because of two main reasons: (1) Calluses do not produce jatrophone. Charlwood, Charlwood & Molina-Torres (1990) point out it is common that dedifferentiated cells lack the capacity for synthesizing isoprenoids present in the mother plant and this is related to its disorganized nature. Similar results are reported in the biosynthesis of heavier terpenes such as steroids (Lui & Staba, 1979). Other possibility is that in vitro cultures may have the ability to synthesize the compounds produced by the field plant but lack the capacity to accumulate them (Banthorpe et al., 1986). In addition, (2) in vitro cultures produce smaller quantities than those detectable by the equipment used (lower than 3 µg/mL). In this regard Pletsch & Charlwood (1997) detected minimum amounts of jatrophone (3 µg/g of dry weight), which were at the detection limit. Nevertheless, cell calluses accumulated a large amount of the compound (1) in relation to that found in the field plant (26 times more) which deserves further studies for identification and biological activity.

In this study, a compound similar to jatrophone in root hexane extracts of J. curcas grown in the field was identified, while the best treatment for callus induction was the addition of 2,4-D 5 µM, together with BAP 2.5 μM in dark conditions. However, the evaluation of the synergy between 2,4-D and NAA is recommended, as well as the evaluation of BAP in concentrations higher than the ones evaluated in the current optimization. Lastly, when evaluating the possible differences between accessions, ARR251107-MFG7 turned out to be the one, which produced the greatest amount of callus. In the analyses of extracts of dedifferentiated cells, desirable compounds of interest, like jatrophone or jatropholones were not detected; however, this approach could be used towards providing large quantities of a particular starting material.

Supplemental Information

Figure S1 Gas chromatogram of an ethanolic extract of Jatropha curcas young callus

A major component (stilbene-like) is shown with retention time of 27.21 min

Click here for additional data file.

Data S1 Analysis of variance

Effect of plant hormones on callus production

Click here for additional data file.

Additional Information and Declarations

Competing Interests

Author Contributions

Data Availability

The authors declare there are no competing interests.

Isidro Ovando-Medina conceived and designed the experiments, wrote the paper, prepared figures and/or tables, reviewed drafts of the paper.

Leny P. Pérez-Díaz performed the experiments, wrote the paper, prepared figures and/or tables.

Sonia Ruiz-González analyzed the data, wrote the paper, reviewed drafts of the paper.

Miguel Salvador-Figueroa conceived and designed the experiments, contributed reagents/materials/analysis tools.

Marcos E. Urbina-Reyes performed the experiments, reviewed drafts of the paper.

Lourdes Adriano-Anaya conceived and designed the experiments, analyzed the data, contributed reagents/materials/analysis tools.

The following information was supplied regarding data availability:

The raw data has been supplied as Supplemental Files.

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
