# Peer review of "Production of cytotoxic compounds in dedifferentiated cells of Jatropha curcas L. (Euphorbiaceae)"

_PeerJ, doi:10.7717/peerj.2616_

## Round 0.1 · original submission · Minor Revisions

It is important that all of the suggested minor amendments to the manuscript are made, especially to english language, and the additional statistical analysis and experimental detail are provided. It is also imperative that the journal format for references and abstract are adhered to.

·

Basic reporting

Abstract does not adhere to journal guidelines, requires headings, refer to ‘standard sections’ of author guidelines
Please review unnecessary use of colon(:) and semi-colon (;) throughout manuscript.
Please confirm that units are expressed as per journal requirements, for example g/mL not gmL-1.
Please address reference list section for adherence to journal format for references, eg. no brackets around publication date, journal names in full (not abbreviated), DOI if known.
Manuscript requires editing by a professional proficient in English to correct sentence structure.

Experimental design

Methods: Please include a reference or protocol used for preparing sulphuric anisaldehyde thin layer chromatography reagent.

Validity of the findings

The findings are presented logically with proper statistical analyses.
Statistical analysis: samples being compared needs to be clarified, appears to be an overuse of subscript letters denoting (in)differences.
Although the original hypothesis was to produce biologically active compounds using callogenesis, the authors have justified appropriately why this was not achieved.
The reporting of negative results are of great value to the broader scientific community.

Additional comments

Specific comments:
Line 42: Possibly substitute the word ‘powerful’ to ‘potent’.
Line 50: Sentence restructuring required. For example “Furthermore, a variety of compounds with different biological activities have been isolated from species belonging to the genus Jatropha”.
Line 78: …dried at 60°C for 48 h……
Line 101: MeOH, not MetOH and throughout manuscript.
Line 115: ‘combined’ not ‘blended’
Line 136: A reference to a compound in brackets (1) is made but there is no structure in the manuscript except as an insert to Figure 3(B). Is this structure (1)? If so please make note of this in figure legend.
Line 163: Please state the yield of the hexane extracts from roots.
Line 232, 235, 236: Keep molar concentration units consistent, for example 6.78 x 10 -6 M should be 6.78 μM etc
Discussion: It could be worth noting that although then callus did not give rise to the desirable compounds of interest, that this approach could be used towards providing large quantities of a particular starting material.
Table 4: Jatropholone b spelt wrong (take out repeated ‘tro’)

·

Basic reporting

1. Minor English language revisions are required as indicated in the PDF.
2. Identification of the stilbene-like compounds will improve the completeness of this publication as this is the major finding in cell culture.
3. References format are not always consistent, should be checked carefully to comply to the Journal requirements.

Experimental design

The objectives of this study were a) to establish a procedure for obtaining friable and
fast growing calluses, and b) to evaluate the production of cytotoxic compounds in J. curcas dedifferentiated cells.

While both objectives are achieved, however the experimental design could be improved as below
1. It has only investigated the callus induction process where the callus was induced and grew well. This is not relevant for the callus been subcultured in the same induction media or indeed very often different media. It is probably more meaningful to investigate the growth of the callus in stable subculture ;
2. In the same context, it is better to investigate the production of cytotoxic compounds in dedifferentiated cells during stable subculture, not in the stage of callus induction.

Validity of the findings

The data is generally robust and statistically sound, and the conclusions are justified.

As stated above, it may be improved by looking into the callus during subcultures where the production of any secondary bioactive compounds would be more meaningful as the callus line would be more stable.

Additional comments

Well-written paper, with some interesting data while not conclusive.

---

## Round 0.2 · accepted · Accept

All minor revisions have been made